# LEPARD: Learning Explicit Part Discovery for 3D Articulated Shape Reconstruction

**Di Liu     Qilong Zhangli     Yunhe Gao     Dimitris N. Metaxas**
Rutgers University

## Abstract

Reconstructing the 3D articulated shape of an animal from a single in-the-wild image is a challenging task. We propose LEPARD, a learning-based framework that discovers semantically meaningful 3D parts and reconstructs 3D shapes in a part-based manner. This is advantageous as 3D parts are robust to pose variations due to articulations and their shape is typically simpler than the overall shape of the object. In our framework, the parts are explicitly represented as parameterized primitive surfaces with global and local deformations in 3D that deform to match the image evidence. We propose a kinematics-inspired optimization to guide each transformation of the primitive deformation given 2D evidence. Similar to recent approaches, LEPARD is only trained using off-the-shelf deep features from DINO and does not require any form of 2D or 3D annotations. Experiments on 3D animal shape reconstruction, demonstrate significant improvement over existing alternatives in terms of both the overall reconstruction performance as well as the ability to discover semantically meaningful and consistent parts.

## 1   Introduction

Predicting the 3D shape and part articulation of an object from a single image is a severely under-constrained and challenging problem. It can be applied to many downstream tasks, such as shape reconstruction [45, 61, 47, 50, 54, 39], segmentation [24, 51, 32, 72, 16, 33, 6, 71], editing [65, 21], re-targeting [12, 20, 69] and medical imaging applications [22, 34, 36, 17, 37, 35, 14, 23, 18, 44]. Successful approaches [26, 29] for predicting the 3D shape of humans rely on a parametric human body model (*e.g.*, SMPL [41]) built from thousands of mocap sequences and on strong supervision from 3D joint locations. Similar breakthroughs are not seen for other articulated object categories, like animals, as 3D scanning of such categories is quite challenging. The lack of 3D annotations and an appropriate parametric animal model has led to approaches that utilize a pre-defined shape template and train with 2D supervision [27, 31, 30, 19, 28]. Assuming such a fixed shape template and supervision from 2D annotations is not optimal, but is necessary for training these systems. Recent works [67, 68, 60] have discarded both assumptions by learning a shape prior (part-based [67, 68] or holistic shape [60]) and by using deep features from an off-the-shelf vision transformer [13], DINO-ViT [5], for supervision. Although the approaches presented in [67, 68, 60] seem promising, they exhibit certain limitations. In [60] a category-specific prior mesh is obtained in a pre-training step, which can be seen as a learned shape template. Then it is trained to predict the articulation and deformation based on this fixed template. LASSIE [67] and Hi-LASSIE [68] learn generic part priors that, similar to [60], can be articulated and deformed. However, learning the deformation field *w.r.t* weak generic shape priors is a hard task. Assuming that the prior shape is far from the target shape of an object, the model has to compensate by predicting large deformations to match the image evidence.

In this paper, we propose a comprehensive framework for reconstructing the 3D shape and related articulations of an object from single-view images that has several desired properties missing from

37th Conference on Neural Information Processing Systems (NeurIPS 2023).

existing works. Similar to [67, 68], the 3D shape of an object is explicitly expressed as a set of part primitives. The primitives are parameterized surfaces (*e.g.*, superquadrics) equipped with additional linear tapering and bending transformations to capture the target shape as faithfully as possible. Unlike [67, 68], the 3D primitives are not fixed across all instances and can deform to capture intra-category variations and accurately reconstruct 3D parts. To capture fine-grained shape details beyond the coverage of the primitive parametric deformations (termed global deformations), we employ a diffeomorphic mapping to estimate local non-rigid deformations of a set of points sampled from the 3D surface of each part. Our approach uses global deformations to capture the salient part of the 3D shape and uses local deformations to further improve the 3D shape reconstruction quality. As such, the local deformations are typically small, which adds to the robustness of our method. Following prior work [66, 68, 60] we use deep features from a vision transformer [13], DINO-ViT [5], as supervision to train our model. Inspired by the kinematics of 3D deformable models [57], we propose a framework to compute image-based forces based on the discrepancy of DINO features and the projected primitive parts. The image-based forces are then converted to generalized forces via kinematics modeling, which provides strong supervision for the deformation of the 3D part primitives.

We conduct extensive experiments on 3D articulated object shape reconstruction through part discovery. The Pascal-part [7] and LASSIE datasets [67] are used for training and evaluation following prior work [67, 68]. Both quantitative and qualitative evaluations demonstrate improved 3D reconstruction performance of our proposed approach compared to existing methods. Our approach even outperforms methods that rely on 3D skeletons or shape templates. In summary, our contributions are as follows:

• A new framework for reconstructing the 3D articulated shape of an object as a set of deformable 3D primitive parts given only 2D evidence. For each primitive global deformations are used to reconstruct the corresponding 3D part, while local deformations increase the fidelity of the reconstruction.

• A kinematics-inspired optimization process with perspective projection that allows converting the 3D primitive points to the generalized parameters corresponding to the transformations of each primitive.

• Extensive quantitative and qualitative evaluations showcase the superiority of our approach over the existing state-of-the-art methods.

## 2 Related Work

**3D reconstruction of animals.** Recently there have been several approaches that learn to reconstruct the 3D shape of animals from image [27, 31, 30, 19, 73, 4, 55, 60] or video inputs [28, 62–64]. Most previous methods make certain assumptions, such as pre-defined shape templates, statistical animal models, or the existence of annotated datasets. For example, some works [73, 4, 55] regress the parameters of a statistical shape model, SMAL [74]. However, this approach is only applicable to categories captured by the SMAL shape space. Some other methods rely on supervision from object silhouettes [27, 31, 30, 19], 2D keypoints [27] and the existence of a template shape [31, 30, 19]. This limits the applicability of those works to animal categories that have such annotations. In contrast, our proposed approach does not require any 3D shape template or skeleton and is trained with a self-supervised objective, which makes it easy to generalize to a wide range of animal species with no extra manual effort.

**Optimization from multiple views and videos.** Recently, Neural Radiance Fields (NeRF) [48, 3] have gained significant attention as a robust volumetric representation for multi-view reconstruction, particularly when accurate cameras are available. In a related line of research, recent works [62–64] have focused on optimizing the 3D shapes of articulated objects using a small number of monocular videos. These approaches employ meticulously designed optimization strategies that incorporate supervision from optical flow, object silhouettes, and DensePose [49] annotations. Another line of work [67, 68], leverages supervision from DINO-ViT [5] features and optimizes a part-based model on a small collection of images ($\sim$ 30) of a particular animal category. LASSIE [67] and Hi-LASSIE [68] additionally include a test-time optimization process, in which per-instance articulation and part-refinement are employed. Our approach is closely related to LASSIE and Hi-LASSIE since we also learn to reconstruct 3D articulated shapes in a part-based manner. However, unlike those methods, our approach does not require any test-time processing.

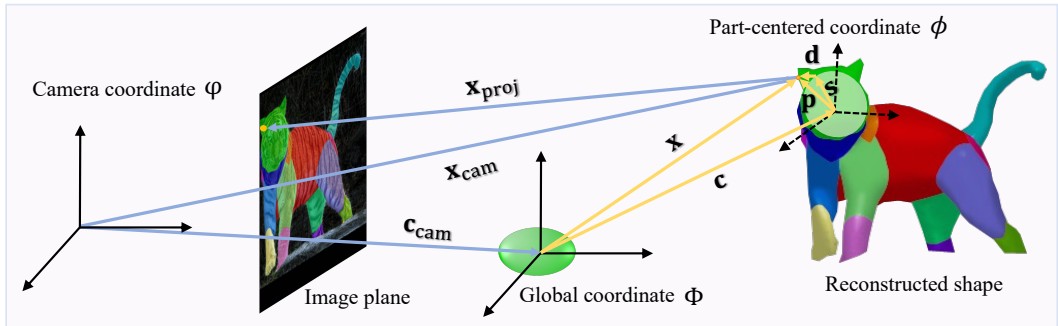

Figure 1: **LEPARD geometry.** We reconstruct an articulated animal shape by deforming a set of primitive parts with global deformation **s** and local deformations **d** that are predicted in a part-centered coordinate system $\phi$. We also predict 3D translation and rotation transformations to place the reconstructed shape in a global coordinate system $\Phi$. Global transformations are parameterized explicitly and offer an intuitive understanding of the shape (*e.g.*, bending parameters make the back and tail of the tiger bend).

**Part discovery.** Deep feature factorization (DFF) [9] and follow-up works [1, 8, 25, 56] show that one can automatically obtain 2D corresponding part segments by clustering deep semantic features. In the 3D domain, the object parts can be discovered by using explicit representation primitives [58, 42, 43, 52, 53, 11] (*e.g.*, cuboids, spheres, superquadrics), or learning part prior [66]. These methods mainly assume some form of supervision like 3D point clouds, keypoints or camera viewpoints. Similar to LASSIE and Hi-LASSIE, we discover parts based on deep features from DINO-ViT. However, we do not rely on a pre-defined 3D skeleton like LASSIE or an intermediate skeleton representation as Hi-LASSIE.

## 3 Approach

### 3.1 Primitive Part Representation

**Geometry.** Given an image of an articulated object category, LEPARD aims to learn a set of $K$ primitive parts that compose its 3D shape. Each primitive is explicitly represented by a group of parameters that describe its 3D shape and orientation. Following [57] each individual primitive $k$ is defined as a closed surface in a part-centered coordinate system $\phi^{(k)}$. Given a point $\mathbf{p}^{(k)}$ on the surface of primitive $k$, its 3D location $\mathbf{x} = (x, y, z)$ in the global coordinate system $\Phi$ can be computed as follows:

$$\mathbf{x} = \mathbf{c}^{(k)} + \mathbf{R}^{(k)}\mathbf{p}^{(k)} = \mathbf{c}^{(k)} + \mathbf{R}^{(k)}(\mathbf{s}^{(k)} + \mathbf{d}^{(k)}), \tag{1}$$

where $\delta^{(k)} \equiv (\mathbf{c}^{(k)}, \mathbf{R}^{(k)})$ represents the transformation of the part-centered coordinate system $\phi^{(k)}$ of primitive $k$ to the global coordinate system $\Phi$, $\mathbf{c}^{(k)} \in \mathbb{R}^3$ and $\mathbf{R}^{(k)} \in \mathbb{R}^{3 \times 3}$ represent the translation and rotation of $\phi^{(k)}$ *w.r.t.* $\Phi$ and $\mathbf{p}^{(k)}$ denotes the relative position of the point on the primitive surface *w.r.t.* $\phi^{(k)}$, which includes global deformation $\mathbf{s}^{(k)}$ and local deformation $\mathbf{d}^{(k)}$. The camera parameters $\pi \equiv (\mathbf{c}_{\text{cam}}, \mathbf{R}_{\text{cam}})$ are used to project $\mathbf{x}$ onto the image. In Fig. 1, we illustrate the geometry of the proposed part-based shape representation that includes global and local deformations for each part. For the sake of simplicity, we omit the superscript $k$ for the $k$-th primitive in the following.

**Primitive deformations.** We employ superquadrics to describe the global deformations of each part primitive. Each superquadric surface $\mathbf{e}$ is explicitly defined by a set of shape-related parameters:

$$\mathbf{e} = a_0 \begin{bmatrix} a_1 \cos^{\varepsilon_1} u \cos^{\varepsilon_2} v \\ a_2 \cos^{\varepsilon_1} u \sin^{\varepsilon_2} v \\ a_3 \sin^{\varepsilon_1} u \end{bmatrix}, \text{where} -\pi/2 \leq u \leq \pi/2, -\pi \leq v \leq \pi. \tag{2}$$

Here, $a_0$ is a scaling parameter, $a_1, a_2, a_3$ denote the aspect ratio for $x$-, $y$-, $z$- axes, respectively, and $\varepsilon_1, \varepsilon_2$ are squareness parameters. To enable more flexible global deformations, we further include tapering and bending parameters. These additional global deformations are defined as continuously

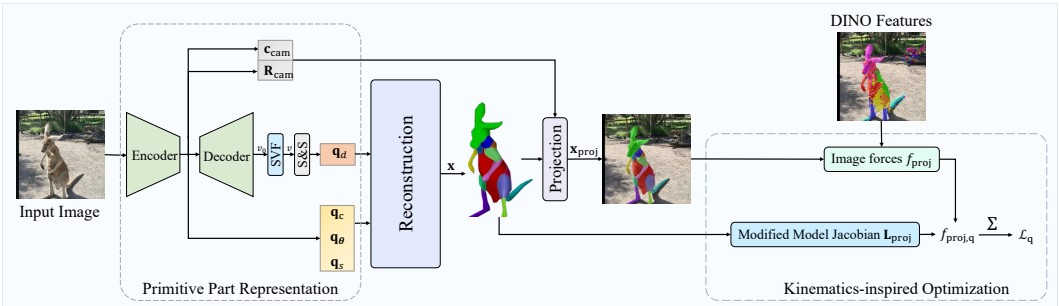

Figure 2: **LEPARD training overview.** Given an input image, we map it to a set of primitive parameters that describe $K$ deformed parts via an encoder-decoder network. The primitive parameters, $\mathbf{q}_c, \mathbf{q}_\theta, \mathbf{q}_s, \mathbf{q}_d$, are then used to reconstruct the 3D articulated shape $\mathbf{x}$. During training, we project a set of points from the primitives' surface onto the image using the predicted camera parameters $\mathbf{c}_{\text{cam}}, \mathbf{R}_{\text{cam}}$. We use DINO features as supervision to compute image-based forces $f_{\text{proj}}$ that we further convert into generalized forces $f_{\text{proj},q}$ to supervise each transformation of the primitive part.

differentiable and commutative functions following [46, 39, 38]. To capture the finer shape details beyond the coverage of global deformations, we employ diffeomorphic point flow to estimate the local non-rigid deformations $\mathbf{d}$. Since the deformation with diffeomorphism is differentiable and invertible [2, 10], it guarantees one-to-one mapping and preserves topology during the non-rigid deformations of the primitives. Please refer to [39, 38] for more details. We note that our approach is not restricted to these types of shapes and respective parameterizations. We can replace it with any differentiable type of primitive. However, for the purposes of the articulated animal shapes this type of parameterization is sufficient.

In summary, each part primitive is represented with a set of parameters $\mathbf{q} = [\mathbf{q}_c, \mathbf{q}_\theta, \mathbf{q}_s, \mathbf{q}_d]$, where $\mathbf{q}_c$ and $\mathbf{q}_\theta$ contain the parameters of the 3D translation and rotation respectively, that transform the part-centered coordinate system $\phi$ of the primitive to the global coordinate system $\Phi$, $\mathbf{q}_s$ are the parameters of the global deformations, $\mathbf{q}_d$ are local deformations that are implemented as a deformation field and are added to the global shape, and $[\cdot]$ is the concatenation operator. Compared to the implicit function-based approaches such as NeRF [48] and NeRS [70], all these primitive parameters are defined explicitly for an intuitive understanding of the primitive deformation.

### 3.2 Primitive 3D Kinematics & Optimization

**Primitive kinematics in 3D.** We use kinematics to define the relationship between any point $\mathbf{x}$ on the primitive surface and the corresponding primitive parameters $\mathbf{q}$. This relationship is expressed quantitatively by the model Jacobian matrix $\mathbf{L}$, and this formulation allows us to deform the primitive shape based on its parameters $\mathbf{q}$. Specifically, the velocity of $\mathbf{x}$ is computed as follows:

$$\dot{\mathbf{x}} = \mathbf{L}\dot{\mathbf{q}}, \tag{3}$$

where $\cdot$ denotes the first-order time derivative and $\mathbf{L} = [\mathbf{I}, \mathbf{B}, \mathbf{RJ}, \mathbf{R}]$ [57] is the Model Jacobian matrix, where each of the four components transforms the 3D points $\mathbf{x}$ into the translation, rotation, global and local deformation parameters of $\mathbf{q}$. $\mathbf{R}$ is the rotation matrix that corresponds to the rotation between the part-centered coordinate system $\phi$ of the primitive and the global coordinate system $\Phi$. $\mathbf{B} = \partial \mathbf{Rp}/\partial \mathbf{q}_\theta$ is related to the rotation matrix $\mathbf{R}$ and the relative position $\mathbf{p}$ of points on the primitive surface. $\mathbf{J} = \partial \mathbf{s}/\partial \mathbf{q}_s$ is the Jacobian matrix. We refer the reader to the supplementary material for more details.

**Optimization in 3D.** In our modeling paradigm, we minimize the energy of the primitive, defined using the principle of virtual work. The 3D forces on the primitive, $f_{\text{3D}}$, result in displacements $d\mathbf{x}$:

$$\mathcal{E}_{f_{\text{3D}}} = \int f_{\text{3D}}^\top d\mathbf{x} = \int f_{\text{3D}}^\top (\mathbf{L}d\mathbf{q}) = \int f_q d\mathbf{q}. \tag{4}$$

where $f_q = f_{\text{3D}}^\top \mathbf{L}$ [57] are the generalized forces acting on the primitive parameters $\mathbf{q}$. $f_{\text{3D}}$ are computed based on the discrepancy between points on the primitive and the target shape in 3D data space (*e.g.*, point-wise difference between the primitive surface and the target shape). The forces on

the primitive are proportional to the distance from the target. Minimizing the generalized forces $f_q$ deforms the primitive to match the target shape and thus can be used as a training objective in our framework.

### 3.3 Primitive Part Discovery from Images

Our model is trained using a set of $N$ in-the-wild images of articulated animals. We do not make use of any type of 2D/3D annotations or skeletons. We compute pseudo-labels using semantic clustering of self-supervised DINO [5] features similar to prior work [67, 68]. For each of the given images, we predict the parameters for the $K$ primitive parts as defined previously. Our approach deforms a set of primitives to fit the target shape under the influence of external forces $f_{3D}$. However, direct primitive parameter optimization using $f_{3D}$ is not feasible since we do not have access to any form of 3D supervision. To this end, we project the predicted 3D primitive parts onto the image space and define losses based on the corresponding 2D forces to supervise the deformation of each primitive. The training overview is given in Fig. 2.

**Projection kinematics from 3D to 2D.** We illustrate the relationship of kinematics between 3D and 2D via projective geometry. This allows us to project the primitives onto the image space and calculate the discrepancy between the projected primitives and 2D evidence. Then, we convert the image forces to their corresponding generalized forces that guide the deformation of the primitives. Specifically, using the estimated parameters for camera translation $\mathbf{c}_{cam}$ and rotation $\mathbf{R}_{cam}$, we can convert a given 3D point $\mathbf{x}$ to the camera coordinate system as follows:

$$\mathbf{x}_{cam} = \mathbf{c}_{cam} + \mathbf{R}_{cam}\mathbf{x}. \tag{5}$$

Under perspective projection, the point $\mathbf{x}_{cam} = (x_c, y_c, z_c)$ projects onto an image point $\mathbf{x}_{proj} = (x_{proj}, y_{proj})$ according to $x_{proj} = f\frac{x_c}{z_c}$, $y_{proj} = f\frac{y_c}{z_c}$, where $f$ is the focal length. By taking the time derivative we have $d\mathbf{x}_{proj} = \mathbf{P}d\mathbf{x}_{cam}$, where

$$\mathbf{P} = \begin{bmatrix} f/z_c & 0 & -fx_c/z_c^2 \\ 0 & f/z_c & -fy_c/z_c^2 \end{bmatrix}. \tag{6}$$

Thus, from Eq. (5) we have:

$$d\mathbf{x}_{proj} = \mathbf{P}d\mathbf{x}_{cam} = \mathbf{P}d(\mathbf{c}_{cam} + \mathbf{R}_{cam}\mathbf{x}) = \mathbf{P}\mathbf{R}_{cam}d\mathbf{x}. \tag{7}$$

Given Eq. (3) we can rewrite Eq. (7) as:

$$d\mathbf{x}_{proj} = \mathbf{P}\mathbf{R}_{cam}d\mathbf{x} = \mathbf{P}\mathbf{R}_{cam}(\mathbf{L}d\mathbf{q}) = (\mathbf{P}\mathbf{R}_{cam}\mathbf{L})d\mathbf{q} = \mathbf{L}_{proj}d\mathbf{q}, \tag{8}$$

where $\mathbf{L}_{proj}$ is the modified model Jacobian matrix that converts image points $\mathbf{x}_{proj}$ into primitive parameters $\mathbf{q}$ that determine translation, rotation, global and local deformations of the primitive.

**Loss components.** Since we only use 2D supervision during training, the primitive energy from each image we minimize is computed based on the image forces $f_{proj}$:

$$\mathcal{E}_{f_{proj}} = \int f_{proj}^{\top}d\mathbf{x}_{proj} = \int f_{proj}^{\top}(\mathbf{L}_{proj}d\mathbf{q}) = \int f_{proj,q}d\mathbf{q}, \tag{9}$$

where the generalized forces $f_{proj,q} = f_{proj}^{\top}\mathbf{L}_{proj}$ guide the primitive deformation. For a given image $i$ from the training dataset, we minimize the discrepancy between our primitives and the target using the image-based forces as follows:

$$\mathcal{L}_q^i = \frac{1}{K}\sum_{k=1}^{K} f_{proj,q}^{(k,i)} = \frac{1}{K}\sum_{k=1}^{K} (f_{proj}^{(k,i)})^{\top}\mathbf{L}_{proj}^{(k,i)}, \tag{10}$$

where $f_{proj}^{(k,i)}$ denotes the corresponding image forces of the $k$-th primitive part and is a vector summation of all the forces at each sampling point on the projected primitive. The final loss used for training is computed by summing the generalized forces for all $N$ training samples as follows:

$$\mathcal{L}_q = f_{proj,q_c}^{\top} + f_{proj,q_\theta}^{\top} + f_{proj,q_s}^{\top} + f_{proj,q_d}^{\top}, \tag{11}$$

where

$$f_{\text{proj},q_c}^\top = \frac{1}{NK} \sum_{i=1}^{N} \sum_{k=1}^{K} (f_{\text{proj}}^{(k,i)})^\top \mathbf{P}^{(k,i)} \mathbf{R}_{\text{cam}}^{(i)}, \qquad f_{\text{proj},q_\theta}^\top = \frac{1}{NK} \sum_{i=1}^{N} \sum_{k=1}^{K} (f_{\text{proj}}^{(k,i)})^\top \mathbf{P}^{(k,i)} \mathbf{R}_{\text{cam}}^{(i)} \mathbf{B}^{(k,i)},$$

$$f_{\text{proj},q_s}^\top = \frac{1}{NK} \sum_{i=1}^{N} \sum_{k=1}^{K} (f_{\text{proj}}^{(k,i)})^\top \mathbf{P}^{(k,i)} \mathbf{R}_{\text{cam}}^{(i)} \mathbf{R}^{(k,i)} \mathbf{J}^{(k,i)}, \quad f_{\text{proj},q_d}^\top = \frac{1}{NK} \sum_{i=1}^{N} \sum_{k=1}^{K} (f_{\text{proj}}^{(k,i)})^\top \mathbf{P}^{(k,i)} \mathbf{R}_{\text{cam}}^{(i)} \mathbf{R}^{(k,i)}.$$

To obtain $f_{\text{proj}}^{(k,i)}$, we render the predicted 3D primitive parts with a differentiable renderer [40] and compute the distance from the pseudo-mask part annotations:

$$f_{\text{proj}}^{(k,i)} = \lambda_{\text{proj}}(\mathcal{G}^{(i)} - \mathcal{M}_{\text{proj}}^{(k,i)}), \tag{12}$$

where $\mathcal{G}^{(i)}$ and $\mathcal{M}_{\text{proj}}^{(k,i)}$ are the pseudo-mask and the $k$-th projected primitive for instance $i$, respectively. $\lambda_{\text{proj}}$ is a constant modeling the strength of the image force $f_{\text{proj}}^{(k,i)}$.

**Force regularization.** To enable more robust primitive fitting strategy, we incorporate regularization to the forces during training. We follow the physics-based deformable models (DMs) [46] and avoid the collisions between primitives by checking for primitive inter-penetration in each training iteration. If two primitives penetrate each other, we allocate two equivalent and opposite collision forces $f_n$ and $-f_n$ that are proportional to the distance between each pair of selected points on the two primitives. These two forces are added to the respective points on the two inter-penetrating primitives, respectively, to adjust the external forces $f_{\text{proj}}$ and thus push the primitives to separate from each other.

## 4 Experiments

**Datasets and baselines.** We emphasize that our aim is to reconstruct an articulated shape by discovering semantically meaningful parts in 3D. Thus, we mainly compare our approach with LASSIE [67] and Hi-LASSIE [68] who share the same goal. We conduct extensive experiments on a few animal categories, including horses, giraffes, zebras, etc. For horses, we train our model using 11k images collected from Pascal [15], LASSIE [67] and DOVE [59] datasets. For the other categories, we only use the images from Pascal [15] and LASSIE [67] datasets to fine-tune our pre-trained model. To put our results into perspective, we also quantitatively compare with A-CSM [30] and 3D Safari [73], which learn to reconstruct the 3D articulated shape of animals holistically. Finally, we also provide qualitative comparisons with LASSIE using their released code. We only compare quantitatively with Hi-LASSIE, using the reported results in [68], since their code is not available at the time of submission.

**Qualitative results.** In Fig. 3 we provide qualitative comparisons with LASSIE. We observe that LEPARD yields more geometrically accurate reconstructions with finer details compared to LASSIE. This is particularly evident in the cases of tiger and elephant, where the primitives used in LASSIE do not accurately correspond to the animal parts with complex shapes such as the tiger tail and the elephant nose. Additionally, LASSIE fails in the presence of additional objects in the background (see the predicted shape for the kangaroo in row 4). For the case of the kangaroo, our model accurately locates the head and correctly recovers the pose under the interference of the inaccurate clustering appearing on the top right of the DINO features. The superiority of our model in estimating accurate poses can also be observed in the case of the penguin (row 5 of Fig. 3).

**Consistency visualization.** In the first two rows of Fig. 4 we show that LEPARD predicts consistent parts across different samples of the same animal category, where LASSIE struggles (*e.g.*, in the 5-th column, LASSIE predicts the elephant's nose as a leg). In addition, we visualize the semantic consistency of our model across different animal categories. In the last two rows of Fig. 4, we compare the reconstruction results of kangaroos and penguins. We observe that our approach employs the same primitive to consistently represent corresponding parts of different species, *e.g.*, the heads of both penguins and kangaroos are represented with the same primitive part (highlighted in green). But LASSIE employs this same green part to represent the mouth of penguins and the head of kangaroos, respectively.

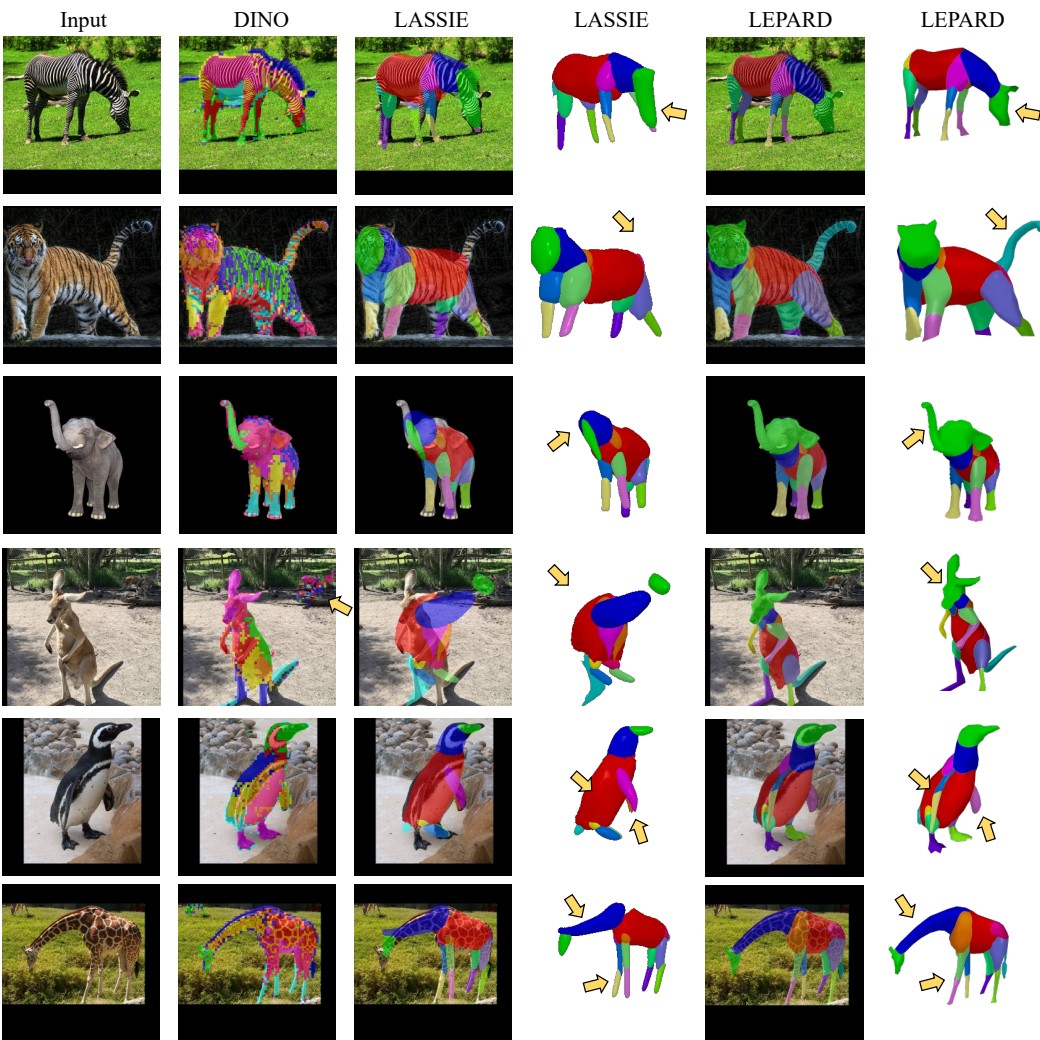

| Input | DINO | LASSIE | LASSIE | LEPARD | LEPARD |
|---|---|---|---|---|---|

Figure 3: **Qualitative results.** We compare the recovered parts by LASSIE [67] and LEPARD (ours). We observe that LEPARD discovers semantically meaningful and consistent parts, offering more faithful reconstructions than LASSIE. Unlike LASSIE, our approach is robust to the presence of additional objects in the background (see the kangaroo in row 4). Failures in LASSIE's prediction are indicated by arrows, whereas LEPARD avoids such failures.

**Keypoint transfer.** Due to the lack of ground-truth 3D annotations in our datasets, we follow a common practice [30, 67, 68] and quantitatively evaluate our model using 2D keypoint transfer between each pair of images in the test set. In particular, given a set of keypoints on a source image, we map them onto the 3D primitive parts and then project them to the target image. We then compute the percentage of correct keypoints (PCK) under a tight threshold $0.05 \times \max(h, w)$ (*i.e.*, PCK@0.05), where $h$ and $w$ are the image height and width respectively. For a successful 2D→3D→2D mapping, accurate 3D reconstructions for both the source and target images are necessary. We report results for the keypoint transfer evaluation in Table 1. The results show that LEPARD achieves higher PCK compared to the baselines without performing test-time per-instance optimization.

**Part transfer.** Next, we evaluate our approach on part transfer using the ground-truth part segmentation masks from the Pascal-Part dataset. Similar to 2D keypoint transfer, we transfer part annotations from source to target images through a 2D→3D→2D mapping utilizing the predicted 3D part primitives for the sources and target images. In this setting, we measure the performance with the percentage of correct pixels (PCP) metric, where a pixel from the source image is considered to be transferred correctly if it is mapped to the same semantic part in the target image. The results

| Input | LASSIE | LEPARD | Input | LASSIE | LEPARD |

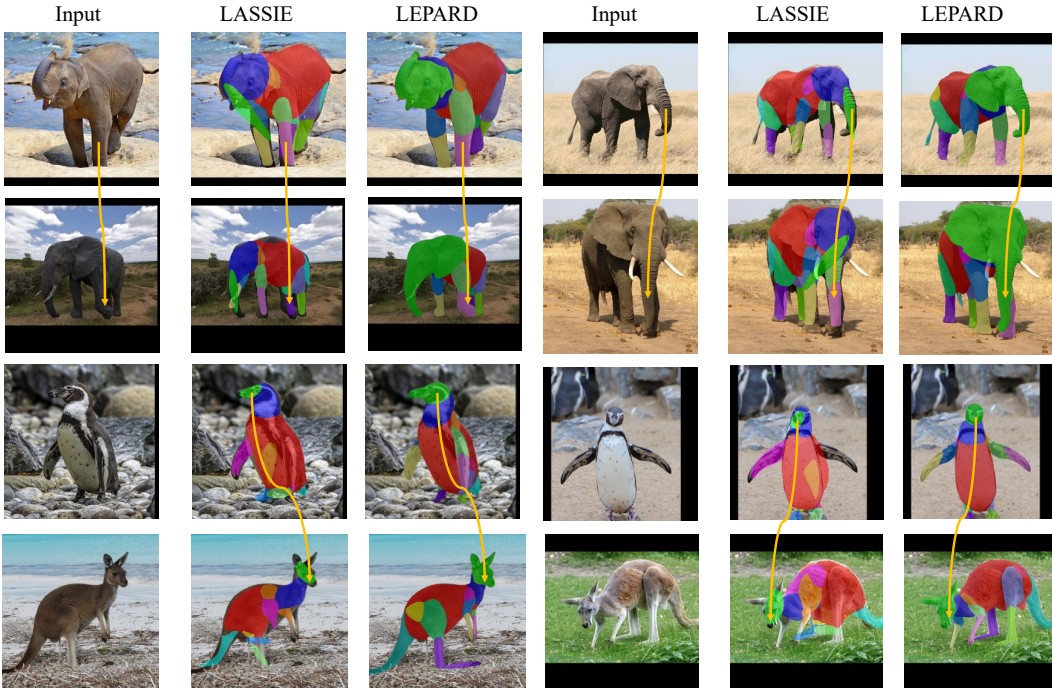

Figure 4: **Consistency visualization.** We compare the discovered parts by LEPARD and LASSIE [67]. The arrows indicate the correct correspondence of the parts among instances (*e.g.*, the left front legs of the elephants should be discovered by primitive parts in the same color) from the same (first two rows) and different (last two rows) animal categories. We observe that LEPARD discovers 3D parts with better semantic consistency (*i.e.*, reconstructs the same parts using the same primitives).

Table 1: **Keypoint transfer evaluation.** Results on the Pascal-Part and LASSIE datasets using PCK@0.05. LEPARD outperforms all methods for all animal categories by a significant margin.

|  | Pascal-Part dataset | | | LASSIE dataset | | | | | |
|---|---|---|---|---|---|---|---|---|---|
|  | Horse | Cow | Sheep | Zebra | Tiger | Giraffe | Elephant | Kangaroo | Penguin |
| 3D Safari [73] | 57.1 | 50.3 | 50.5 | 62.1 | 50.3 | 32.5 | 29.9 | 20.7 | 28.9 |
| A-CSM [30] | 55.3 | 60.5 | 54.7 | 60.3 | 55.7 | 52.2 | 39.5 | 26.9 | 33.0 |
| LASSIE [67] | 58.0 | 62.4 | 55.5 | 63.3 | 62.4 | 60.5 | 40.3 | 31.5 | 40.6 |
| Hi-LASSIE [68] | 59.6 | 63.1 | 56.2 | 64.2 | 63.1 | 61.6 | 42.7 | 35.0 | 44.4 |
| LEPARD | **61.0** | **63.7** | **56.9** | **64.7** | **63.8** | **62.1** | **43.2** | **35.4** | **44.6** |

are reported on the right column of Table 2, demonstrating that LEPARD compares favorably to all baseline methods.

**2D IoU.** In addition, we evaluate LEPARD using overall and part IoU between the ground-truth and rendered masks. The results are reported in Table 2. We follow the baselines [68, 67, 25, 5] and manually assign the discovered parts to the best-matched parts in the Pascal-Part segmentation masks. From Table 2 we observe that LEPARD outperforms all other methods in terms of overall IoU as well as Part IoU by a large margin.

**Effect of local deformations.** We investigate the effect of local deformations in terms of reconstruction accuracy and present qualitative results in Fig. 5. We observe that local deformations can capture fine-grained shape details and significantly improve the visual quality of the reconstructed shapes.

**Model robustness and limitation.** In Fig. 6 we evaluate on some held-out images from the Objaverse dataset for known categories such as elephants to show the robustness of our method. For the reconstruction of unknown categories, our method may require fine tuning on additional images, and will be included in our future work.

Table 2: **Quantitative results on the Pascal-Part [7] dataset.** We report the overall IoU, part mask IoU, as well as part transfer results measured by the percentage of correct pixels (PCP).

| | Overall IoU | | | Part IoU | | | Part Transfer (PCP) | | |
|---|---|---|---|---|---|---|---|---|---|
| | Horse | Cow | Sheep | Horse | Cow | Sheep | Horse | Cow | Sheep |
| SCOPS [25] | 62.9 | 67.7 | 63.2 | 23.0 | 19.1 | 26.8 | - | - | - |
| DINO clustering [1] | 81.3 | 85.1 | 83.9 | 26.3 | 21.8 | 30.8 | - | - | - |
| 3D Safari [73] | 72.2 | 71.3 | 70.8 | - | - | - | 71.7 | 69.0 | 69.3 |
| A-CSM [30] | 72.5 | 73.4 | 71.9 | - | - | - | 73.8 | 71.1 | 72.5 |
| LASSIE [67] | 81.9 | 87.1 | 85.5 | 38.2 | 35.1 | 43.7 | 78.5 | 77.0 | 74.3 |
| Hi-LASSIE [68] | 83.4 | 88.1 | 86.3 | 39.0 | 35.3 | 43.4 | 79.9 | **77.8** | 75.5 |
| LEPARD | **83.7** | **88.3** | **86.7** | **39.5** | **35.4** | **43.6** | **80.6** | 77.6 | **75.8** |

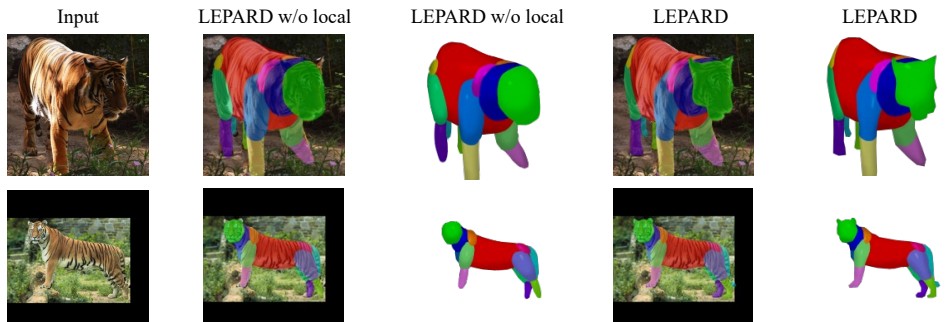

Figure 5: **Effect of local deformations.** We test the effectiveness of the local deformations on the Pascal-Part and the LASSIE dataset. We observe that local deformations have a significant impact on the visual quality of the reconstructed shapes.

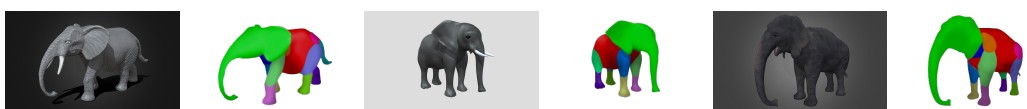

Figure 6: Testing on held-out images from known category. We show some qualitative results on the elephant category of Objaverse dataset.

## 5 Conclusion

We present LEPARD, a learning-based framework for discovering the 3D parts and shape of an articulated animal from an image without any 2D/3D annotations or skeletons. Our approach jointly optimizes a set of primitives which are explicitly defined by a few shape-related parameters, and provides an intuitive understanding of the primitive deformation. To obtain high-fidelity reconstruction that matches the 2D evidence provided by clustering DINO features, we propose a kinematics mapping between 3D and 2D, and convert the 2D forces defined *w.r.t.* images to the generalized forces that supervise the motions and deformations of the primitive parts in 3D. Our approach does not require any test-time processing and discovers semantically meaningful and consistent 3D parts. We demonstrate the effectiveness of our approach through quantitative and qualitative evaluations where we achieve significant improvements over existing methods.

**Acknowledgments**

This research has been partially funded by research grants to D. Metaxas through NSF: IUCRC CARTA 1747778, 2235405, 2212301, 1951890, 2003874, and NIH-5R01HL127661.

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
