# Supplementary Material for LEPARD: Learning Explicit Part Discovery for 3D Articulated Shape Reconstruction

**Di Liu**    **Qilong Zhangli**    **Yunhe Gao**    **Dimitris N. Metaxas**
Rutgers University

In this supplementary material, we provide additional details and results that were not included in the main paper due to space constraints. In Sec. A we provide additional details for the formulation of our proposed approach. Next, in Sec. B we present implementation details. Finally, in Sec. C we present additional qualitative results.

## A   Additional Formulation Details

### A.1   Notation table

Table 3: **Notations.** We list the symbol, variable name, state space, and some notes for the key variables in the paper.

| Symbol | Variable name | State space | Notes |
|---|---|---|---|
| $\mathbf{c}_{\text{cam}}$ | Camera translation | $\mathbb{R}^3$ | |
| $\mathbf{c}$ | Primitive translation | $\mathbb{R}^3$ | |
| $\mathbf{R}$ | Primitive rotation | $SO(3)$ | |
| $\mathbf{R}_{\text{cam}}$ | Camera rotation | $SO(3)$ | |
| $\mathbf{s}$ | Global deformations | $\mathbb{R}^3$ | |
| $\mathbf{e}$ | Superquadric surface | $\mathbb{R}^3$ | |
| $\mathbf{d}$ | Local deformations | $\mathbb{R}^N$ | $N$: sampling points on primitive surface |
| $\mathbf{q}_c$ | Parameters for primitive translation | $\mathbb{R}^3$ | $\mathbf{q}_c = \mathbf{c}$ |
| $\mathbf{q}_\theta$ | Parameters for primitive rotation | $\mathbb{R}^4$ | $\mathbf{q}_\theta$ is a 4D quaternion vector following [3, 8, 7] |
| $\mathbf{q}_s$ | Parameters for global deformations | $\mathbb{R}^{11}$ | $\mathbf{q}_s = (\mathbf{a}, \varepsilon, \mathbf{t}, \mathbf{b}), \mathbf{a} \in \mathbb{R}^4, \varepsilon \in \mathbb{R}^2, \mathbf{t} \in \mathbb{R}^2, \mathbf{b} \in \mathbb{R}^3$ |
| $\mathbf{q}_d$ | Parameters for local deformations | $\mathbb{R}^N$ | $\mathbf{q}_d = \mathbf{d}$ as we use one to one mapping |
| $\mathbf{B}$ | Rotation related matrix | $\mathbb{R}^{3 \times 4}$ | $\mathbf{B} = \partial\mathbf{R}\mathbf{p}/\partial\mathbf{q}_\theta$ |
| $\mathbf{J}$ | Jacobian matrix | $\mathbb{R}^{3 \times 11}$ | $\mathbf{J} = \partial\mathbf{s}/\partial\mathbf{q}_s, \mathbf{J} \in \mathbb{R}^{3 \times 6}$ if no global deformations |

### A.2   Primitive kinematics in 3D

In this section, we provide detailed derivation for the kinematics proposed in the main paper. Specifically, given a point $\mathbf{p}$ on the primitive surface, its 3D location $\mathbf{x} = (x, y, z)$ in the global coordinate system $\Phi$ is

$$\mathbf{x} = \mathbf{c} + \mathbf{R}\mathbf{p} = \mathbf{c} + \mathbf{R}(\mathbf{s} + \mathbf{d}). \tag{15}$$

From Eq. (15), we can derive the velocity of a point on the primitive surface as

$$\dot{\mathbf{x}} = \dot{\mathbf{c}} + \dot{\mathbf{R}}\mathbf{p} + \mathbf{R}\dot{\mathbf{p}} = \dot{\mathbf{c}} + \mathbf{B}\dot{\mathbf{q}}_\theta + \mathbf{R}\dot{\mathbf{s}} + \mathbf{R}\mathbf{S}\dot{\mathbf{q}}_d, \tag{16}$$

where $\cdot$ denotes the first-order time derivative. $\mathbf{B} = \partial\mathbf{R}\mathbf{p}/\partial\mathbf{q}_\theta$ is a $3 \times 4$ transformation matrix related to the rotation matrix $\mathbf{R}$ and the relative position $\mathbf{p}$ of points on the primitive surface. $\dot{\mathbf{s}} = [\partial\mathbf{s}/\partial\mathbf{q}_s]\dot{\mathbf{q}}_s = \mathbf{J}\dot{\mathbf{q}}_s$, where $\mathbf{J}$ is the Jacobian matrix of the model-centered coordinates $\phi$ *w.r.t.* the global deformation parameters at each point. We note that the size of the Jacobian matrix is determined

37th Conference on Neural Information Processing Systems (NeurIPS 2023).

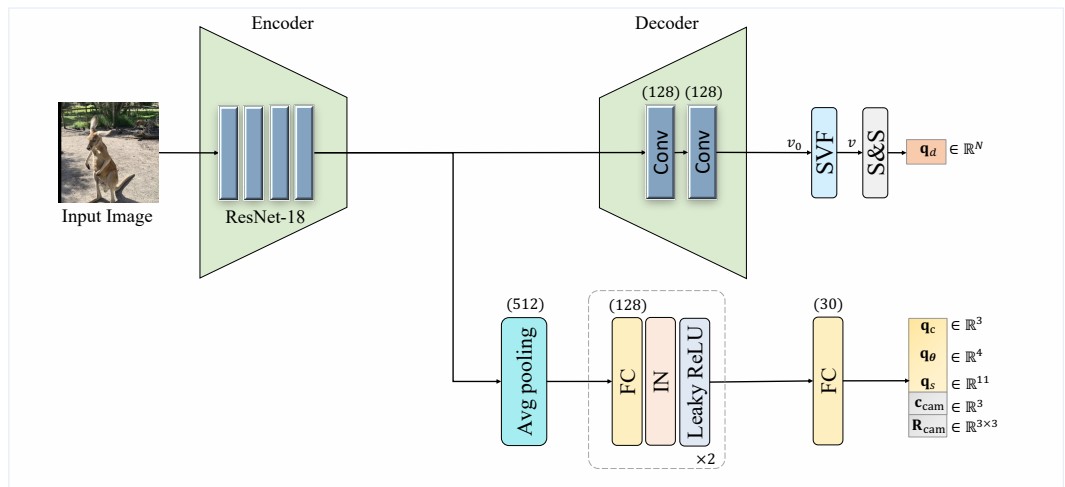

Figure 8: **Network architecture.** We show the architecture of the encoder-decoder for the estimation of primitive parameters. The numbers in (·) indicate the dimension of output features.

by the type of global deformations used. $\mathbf{S}$ is a shape matrix that we set to the identity matrix $\mathbf{I}$ in LEPARD since we use one-to-one mapping for the local deformation estimation. Eq. (16) can be further written in the form:

$$\dot{\mathbf{x}} = [\mathbf{I}, \mathbf{B}, \mathbf{RJ}, \mathbf{R}]\dot{\mathbf{q}} = \mathbf{L}\dot{\mathbf{q}}, \tag{17}$$

where $\mathbf{L}$ is the overall deformable model's Jacobian matrix (termed model Jacobina matrix) that includes the Jacobians for translation, rotation, global and local deformations [6]. Eq. (17) shows the relationship between any point $\mathbf{x}$ on the 3D primitive surface and its corresponding primitive parameters $\mathbf{q}$ that control the transformation of the primitive. As shown in Sec. 3.2 of the main paper, this kinematic formulation further allows us to convert the 3D forces $f_{3D}$ to the generalized forces $f_q$ which we use to supervise the primitive transformation during training.

## B    Implementation Details

### B.1    Training protocol

We implement the LEPARD training using PyTorch on eight Nvidia A100 GPUs and optimize all network parameters using an Adam optimizer [5]. To extract the semantic features from images, we follow [9, 10] and use a self-supervised ViT (DINO-ViT) which is trained using a self-distillation approach [1]. Specifically, we extract the keys from the last layer of DINO given an input image with the size $512 \times 512$ and obtain a feature map with size $64 \times 64$. Similarly, we extract the class tokens and use their average attention map as a saliency estimation. We then collect and cluster the features of salient image patches by thresholding the saliency scores. The feature clustering is done by an off-the-shelf K-means algorithm with four clusters. Finally, we obtain a pseudo ground-truth object silhouette $\mathcal{G}$ by thresholding the minimum feature distance to the center of the clusters.

### B.2    Network architecture

In Fig. 8, we provide the architecture of the encoder-decoder model proposed in the main paper. The architecture comprises two main components: 1) a feature encoder to map the input image into a low-dimensional feature map that is further used to output a set of global primitive parameters and 2) a decoder with convolutional layers and a diffeomorphic mapping to predict the local deformation parameters.

In our experiments, the feature encoder is a ResNet-18 [4] that is pre-trained on ImageNet [2]. From the original architecture, we remove the final fully connected layer and keep only the feature vector of length 512 after global average pooling. Subsequently, we use two MLP modules and a fully connected (FC) layer to map this to the global primitive parameters $\mathbf{c}_{cam}, \mathbf{R}_{cam}, \mathbf{q}_c, \mathbf{q}_\theta, \mathbf{q}_s$ that correspond to camera translation and rotation, primitive translation and rotation, and global

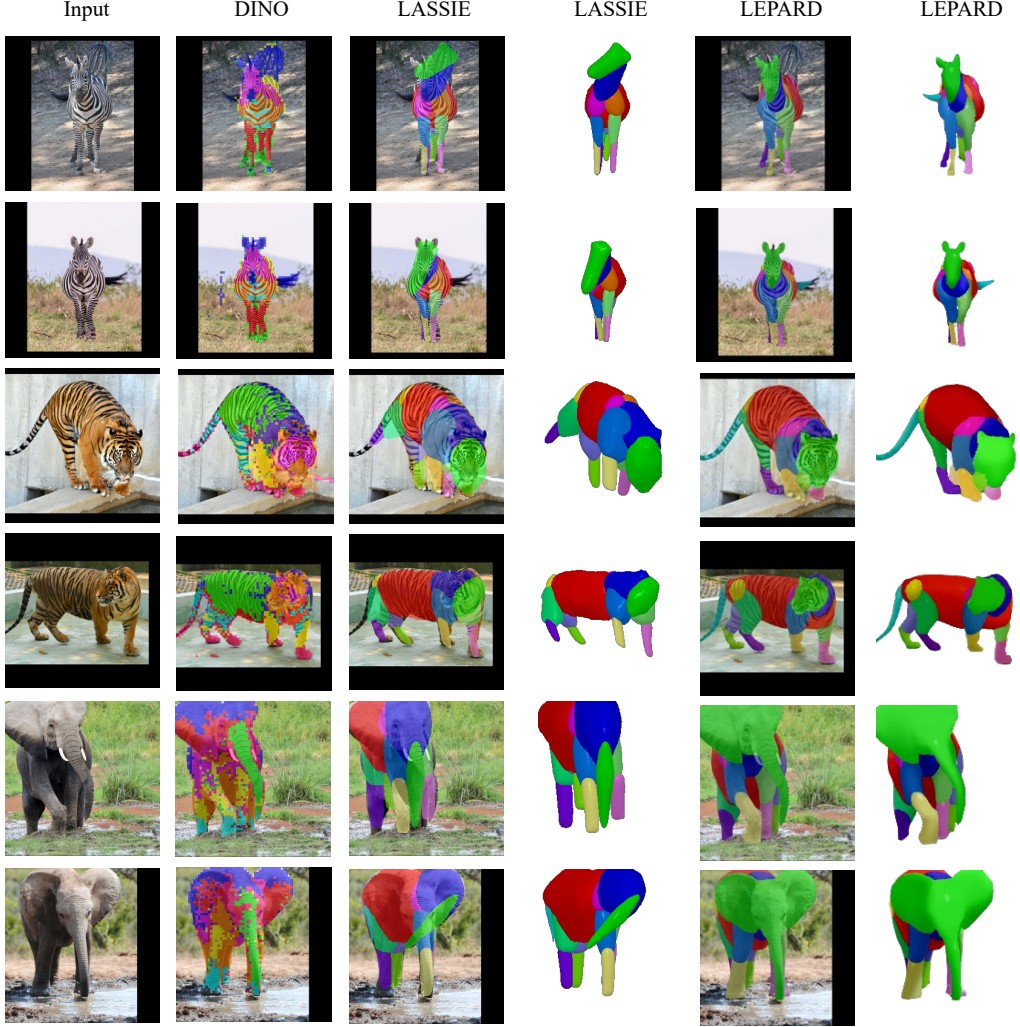

| Input | DINO | LASSIE | LASSIE | LEPARD | LEPARD |

Figure 9: **LEPARD** *v.s.* **LASSIE**. We compare LEPARD with LASSIE and show part discovery results on their self-collected animal image ensembles.

deformations of each primitive part. Each MLP module consists of a fully connected layer with 128 hidden dimensions, an instance normalization, and a Leaky ReLU activation. During training, we use the pre-trained batch statistics for normalization.

The decoder comprises two convolutional layers with an output size of 128 to map the encoded feature to a vector filed $v_0$ which is further used as the input of the diffeomorphic mapping. We employ a Gaussian smoothing layer to map $v_0$ to a stationary velocity field (SVF) $v$ and obtain the local deformation parameters $\mathbf{q}_d \in \mathbb{R}^N$ using a scaling and squaring (S&S) layer. We uniformly sample $N =$1K points for each primitive part during training.

## C   Additional Results

**Additional comparisons with LASSIE.** We show more qualitative results on various animal categories in Fig. 9 and Fig. 10, comparing our approach with LASSIE. As can be seen in Figs 9 & 10, LEPARD yields higher quality reconstructions than LASSIE.

**Comparison with Hi-LASSIE.** In addition to the quantitative comparison with Hi-LASSIE [10] using their reported numbers in the main paper, in Fig. 11 we provide a visual comparison using some sample results taken from their paper. We observe that our approach generates more accurate

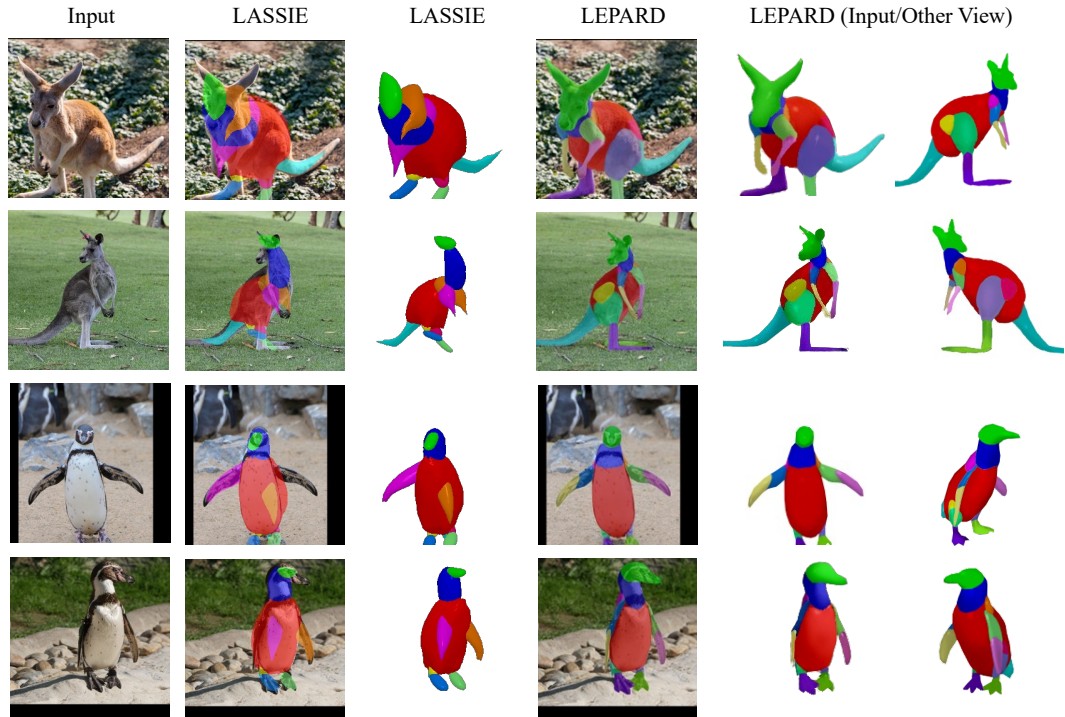

Figure 10: **LEPARD *v.s.* LASSIE**. We compare LEPARD with LASSIE and show part discovery results on their self-collected animal image ensembles.

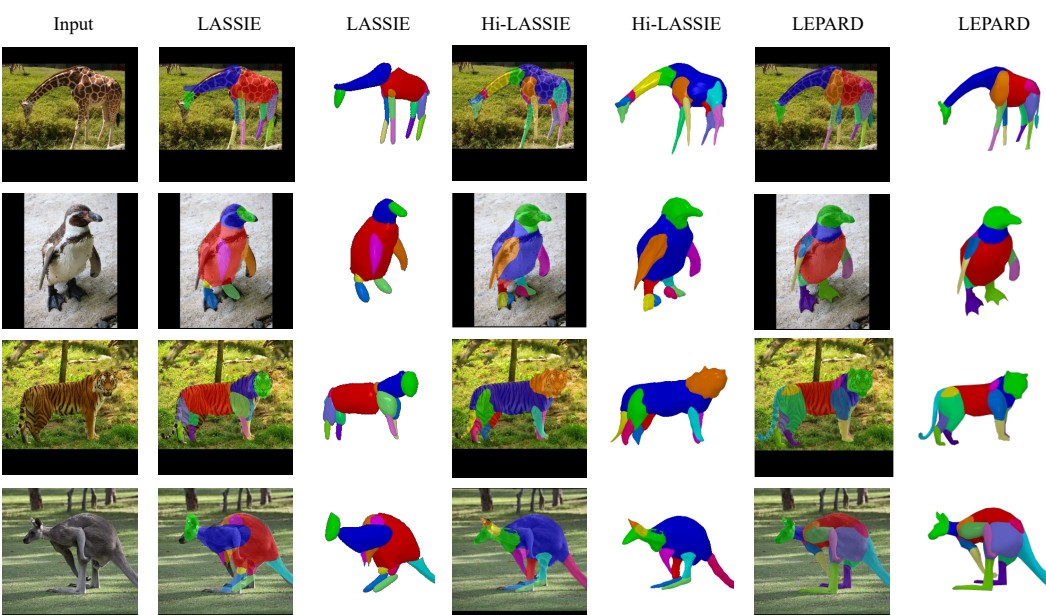

Figure 11: **LEPARD *v.s.* Hi-LASSIE**. We provide results of LEPARD compared to LASSIE [9] and Hi-LASSIE [10]. The visual results of Hi-LASSIE are taken from [10] as they haven't released their code.

3D articulated shapes with finer details than LASSIE and Hi-LASSIE. Moreover, our approach can preserve the semantic consistency among different species, while Hi-LASSIE uses a variant number of primitive parts with sub-optimal semantic meaning to reconstruct the shapes of different animal categories.