# OpenReview forum: "LEPARD: Learning Explicit Part Discovery for 3D Articulated Shape Reconstruction"
_NeurIPS.cc/2023/Conference — NeurIPS 2023 poster_

### Official Review · Reviewer_Hc7X · 2023-06-27

**Soundness:** 2 fair
**Presentation:** 2 fair
**Contribution:** 3 good
**Rating:** 4
**Confidence:** 2

**Summary:**

The paper studies the research problem of shelf(DINO)-supervised articulated 3D shape reconstruction. The key idea of the paper is to factorize shapes into different primitives, and to model the shape primitives using both global and local deformation parameters. The parameters of the model are optimized in a kinematics-inspired way, where the 3D kinematics are projected into 2D to utilize the DINO supervision. The experiments show the proposed model outperforms previous state-of-the-art methods, including both primitive and holistic reconstruction approaches.

**Strengths:**

1. The high-level idea of factorizing primitive parameterization into global and local deformations make sense.
2. The performance of the proposed method has been well evaluated and clearly outperforms previous state-of-the-art methods.

**Weaknesses:**

1. As a non-expert in this field who does not have much knowledge about physics-based deformable model, I find this paper hard to follow and sometimes not self-contained. Not being able to understand some details about kinematics could be caused by my limited knowledge about physics, but, importantly, it's not clear to me at all why the kinematics-based optimization approach is adopted in this paper. Conceptually, I understand optimizing primitives without direct 3D primitive supervision requires some sort of regularization -- but what kind of prior knowledge or physical facts are encoded in this model, given that there are no real forces/materials? What are the assumptions/hypothesis that generate these forces? And what makes the local deformation to be small/local? It is really important to make the high-level intuition clear before dive into the details. I also wonder if one uses the global+local deformable model in this paper but with simple optimization methods (e.g. directly sample points from primitive surface and project to 2D, then minimize the mask L1/MSE/BCE loss), will such simple alternatives not converge/lead to inferior performance?

2. In the quantitative ablation (Fig.6), the performance gap between the model w/o local and the full model does not seem too large (e.g. compared to the gap between LEPARD and Hi-LASSIE in Table 2). What is the typical std/error bar of this evaluation?

Other comments/questions:
1. It would have been beneficial to provide the dataset statistics.
2. How is K decided? (L164)
3. What is the limitation/failure cases of the proposed method?

**Questions:**

Please see the questions in the weaknesses. Overall, the high-level idea of global+local deformations modeling makes sense, and the qualitative results of the proposed method looks impressive. However, I do think the writing of the paper needs major improvement and many details need further clarification.

**Limitations:**

A detailed discussion about limitation and failure case is missing now and should be included.

---

> ### Author Rebuttal · Authors · 2023-08-09
>
> >Q1: Motivation of kinematics-based optimization approach adopted in this paper.
>
> A1: Thanks for the comment! We realize that we may have made presumptions regarding the reader's familiarity with PDMs, and we appreciate the feedback. Let's break down the approach and the intuition behind our choices:
>
> **Virtual Forces**: The forces we reference in the paper aren't real forces we encounter in physical systems. Instead, these are "virtual forces". They're computed based on the virtual displacement of the surface points of the primitive[33]. They are used to measure how well the points on the primitive surface are deformed to match the target shape.
>
> **Kinematic-Based Optimization**: The choice to employ kinematic-based optimization stems from the conceptualization of the primitive deformation as a Lagrangian dynamic system[33]. This means the points on the primitive surface are continuously deforming to match the target shape during the training iterations, and also the shape-related parameters of the primitive change accordingly. During each training iteration, by minimizing the virtual forces applied to the primitives, we can obtain a reconstructed shape that is closest to the target.  However, we need more effective regularization due to the lack of 3D supervision. In addition to directly sampling points from the primitive surface and optimizing them (which might seem straightforward),  we seek to optimize groups of shape-related parameters that control the transformations of the primitive, i.e., translation, rotation, global and local deformations. This imposes tighter constraints, and is crucial especially when we do not have any 3D supervision but only 2D image-based evidence. Our method essentially provides more constrained regularization, making the model more robust and accurate in the absence of 3D supervision. We will carefully revise the paper by simplifying the text and adding intuition to make it easier to read.
>
> >Q2:  I also wonder if one uses the global+local deformable model in this paper but with simple optimization methods (e.g. directly sample points from the primitive surface and project to 2D, then minimize the mask L1/MSE/BCE loss), will such simple alternatives not converge/lead to inferior performance?
>
> A2: We understand your concern, and added an ablation study using our primitive parameterization with only image force loss (MSE). The results are given in Table. 2 of the Author Rebuttal PDF.
>
> >Q3: In the quantitative ablation (Fig.6), the performance gap between the model w/o local and the full model does not seem too large (e.g. compared to the gap between LEPARD and Hi-LASSIE in Table 2). What is the typical std/error bar of this evaluation?
>
> A3: The local deformations capture the fine-grained shape details and greatly improve the visual quality of the reconstructed shapes, while their effect is not entirely captured by the quantitative metrics that only improve slightly. We also added a std/err bar in Table. 2 of the Author Rebuttal PDF.
>
> >Q4: It would have been beneficial to provide the dataset statistics.
>
> A4: We train our model on Pascal-Part and LASSIE dataset following[35,40], and use the same evaluation protocol as LASSIE.
>
> >Q5: How is $K$ decided? Q6: Limitation
>
> A5, A6: Please see the general Author Rebuttal for details.

---

> > ### Comment · Reviewer_Hc7X · 2023-08-13
> >
> > I thank the authors for their detailed response. Although I still don't fully understand what the "tighter constraints"/"constrained regularization" is intuitively (I strongly encourage the authors to spend more time on this for a broader audience), my other concerns have been addressed. I am willing to raise my rating (still with low confidence though).

---

> > > ### Author Response · Authors · 2023-08-16
> > > **Response to Reviewer Hc7X**
> > >
> > > Thank you for your constructive feedback! We will carefully revise our paper for better readability. We genuinely appreciate your willingness to reconsider the rating based on our explanations. We'd be thankful if you could take a moment to update the rating in the system. Let us know if you have any questions!

---

### Official Review · Reviewer_a5TN · 2023-07-02

**Soundness:** 2 fair
**Presentation:** 1 poor
**Contribution:** 2 fair
**Rating:** 5
**Confidence:** 3

**Summary:**

The paper introduces LEPARD, a framework for reconstructing the 3D shape of animals from single images. LEPARD reconstructs 3D shapes as parts, which are parameterized primitive surfaces with global and local deformations. LEPARD is trained using off-the-shelf deep features without the need for 2D or 3D annotations. Experimental results demonstrate more detailed shape reconstruction.

**Strengths:**

**Method:**
- 3D reconstruction as parts is an interesting approach to 3D reconstruction and is generally less explored than holistic reconstruction. Exploration in this direction can be beneficial to the community.
- The proposed method reconstructs details better by modeling coarse shapes as well as detailed deformations of each part. This is an intuitive approach and appears to be effective.

**Experiments:**
- The comparison with SOTA is thorough and complete to the best of my knowledge. The improvements over SOTA appear to be significant.
- The ablation study clearly demonstrates the effectiveness of each component.

**Weaknesses:**

I have several confusion about the method.

- The model parameters are not defined precisely. It’s unclear how q_c and q_theta are defined ( how are they just c and R? or is there any difference?)
- Details regarding how the local deformation d is obtained from v_s are missing. The concept of stationary velocity field and Gaussian smoothing layer should be explained more in detail to help understanding. Also, the paper should include justification for using stationary velocity field and Gaussian smoothing layer.
- I don’t understand Equation 5. The author introduces the model Jacobian matrix without defining what the model is. It’s also unclear why the velocity x is relevant in this static reconstruction setting. In addition, it’s unclear how Equation 5 is derived.
- In L152-153, what is the practical meaning of “the energy of the primitive” and how is this energy and the force f_3d energy related to 3D shape reconstruction?
- how is the loss in equation 12 derived?
- In equation 14, shouldn’t G^i be part-based (as described in L162 and illustrated in Fig 3) and hence be denoted as G^(i,k)? Otherwise, why is the rendered part mask compared with the mask of the whole object? If G^i is part-based, how are the part labels from the DINO feature associated with the K primitives?

I find it hard to understand the proposed method and hence cannot recommend acceptance given the current state. Please address my questions above

**Questions:**

Questions are listed in weaknesses.

**Limitations:**

There is no discussion about limitations in the paper.

---

> ### Author Rebuttal · Authors · 2023-08-09
>
> Thanks for your detailed comments! We will release the code for reproduction. We hope the following responses can address your concerns.
>
> >Q1: The model parameters are not defined precisely. It’s unclear how $q_c$ and $q_\theta$ are defined.
>
> A1: $q_c = c \in \mathbb R ^3$ represents the 3D translation of each part, while $q_\theta  \in \mathbb R ^4$ is a standard 4D quaternion formulation that represents the 3D rotation of each part. Please also see L134-135 of the main paper and the Notes in the Notation table (Table 3 of the supplementary material).
>
> >Q2: Details regarding how the local deformation d is obtained from $v$ are missing. The concept of SVF and Gaussian smoothing layer should be explained more in detail to help understanding.
>
> A2: The local deformation $\textbf{d}$ is modeled as diffeomorphic point flows, which is a smooth and invertible spatial transformation (diffeomorphic mapping) and is widely used in shape modeling[1,2]. Diffeomorphic mappings allow for the computation of a time-dependent velocity field via an ordinary differential equation (ODE). However, such computations can be intricate and resource-intensive. To address this, a stationary velocity field (SVF) is usually employed [3]. This SVF maintains a constant velocity and simplifies the parameterization of diffeomorphisms. More specifically, a diffeomorphic mapping is achieved as the trajectory and integration of a smooth SVF $v$. The widely accepted practice[2] employs a scaling and squaring layer (S&S) for the integration, and utilizes convolution with suitable kernels, e.g., Gaussian kernels with positive scale,  to smooth out the velocity ﬁelds and achieve a more refined output - this process is termed the Gaussian smoothing layer. More comprehensive explanations of this can be found in Section A.3 of the supplementary.
>
> [1] Beg, M. Faisal, et al. "Computing large deformation metric mappings via geodesic flows of diffeomorphisms." IJCV, 2005.
>
> [2] Amor, Boulbaba Ben, Sylvain Arguillère, and Ling Shao. "ResNet-LDDMM: advancing the LDDMM framework using deep residual networks." TPAMI, 2022.
>
> [3] Arsigny, Vincent, et al. "A log-euclidean framework for statistics on diffeomorphisms." MICCAI, 2006.
>
> >Q3: I don’t understand Equation 5. The author introduces the model Jacobian matrix without defining what the model is. It’s also unclear why the velocity x is relevant in this static reconstruction setting. In addition, it’s unclear how Equation 5 is derived.
>
> A3: Equation 5 shows how a change in a 3D point $x$ is translated to a change in the shape-related parameters of the primitive. Our approach is inspired by the kinematics of physics-based deformable models [33], which take the shape deformation as a dynamic system and thus the time derivative is used for $x$ and $q$. In our setting, at an arbitrary training iteration $t$, the 3D point $x$ from the primitive surface should match with the $x_{gt}$ point from the surface of the ground-truth shape. $\dot{x}$ a.k.a $d{x}$ shows how this point should change to make this possible and $\dot{q}$ a.k.a $dq$ is the corresponding change in the shape-related parameters of the primitive. Thus, in the static reconstruction setting the model is deformed during training to match the ground-truth shape. Please see Sec. A.4 of the supplementary material for detailed derivation of Equation 5.
>
> >Q4: In L152-153, what is the practical meaning of “the energy of the primitive” and how is this energy and the force $f_{3d}$ energy related to 3D shape reconstruction?
>
> A4: In the context of 3D shape reconstruction, the "energy of the primitive" refers to the amount of virtual work required to deform a primitive so that it aligns with a target shape. Here, "virtual work" is defined as the product of the virtual force and the distance over which this force acts, specifically the distance between the primitive surface and the target shape. During the training process, the goal is to minimize this energy, ensuring that the primitive shape deforms and closely matches the target shape. In other words, the lesser the energy, the closer our primitive is to the desired 3D shape.
>
> >Q5: how is the loss in equation 12 derived?
>
> A5: As we explained in Q4, we try to minimize the primitive energy to deform the primitive to match the target shape. The primitive energy refers to the amount of virtual work defined as the integration of “force” $f_\text{proj}$  over the change of the points $dx_\text{proj}$ (Eq. 11). Using Eq. 5, the change of points $dx_\text{proj}$ is further expressed as the product of Jacobian $L_\text{proj}$ and change of shape-related parameters $d q$. Thus we can convert the minimization of 2D point-wise image forces $f_\text{proj}$ to the minimization of the parameter-based forces (generalized forces) $f_\text{proj}$ using the Jacobian matrix, which leads to Eq. 12.
>
> >Q6: In equation 14, shouldn’t $G^i$ be part-based and hence be denoted as $G^{(i,k)}$?
>
> A6: Since the semantic clusters from DINO features do not essentially match the number of parts, we do not use the part-based ground truth but use the whole mask. During training, all primitive parts are initialized and deformed together to reach the boundary of the mask. To avoid the intersections between primitives, we follow PDMs and check for primitive inter-penetration in each training iteration. If two primitives penetrate each other, we assign two equivalent and opposite collision forces $f_n$ and $-f_n$ that are proportional to the distance between each pair of selected points on the two primitives. These two forces are added to the respective points on the two inter-penetrating primitives, respectively, to adjust the forces $f_\text{proj}$ and thus push the primitives to separate from each other. In this way, the primitives are deformed to different parts of the shape under the kinematic-based optimization. We will add all the necessary details to the revised paper/supplementary.

---

> > ### Comment · Reviewer_a5TN · 2023-08-15
> >
> > Thanks for their detailed response and for the clarifications. My questions are properly addressed. It would be helpful to include the clarifications in the final version. I am willing to raise my rating.

---

> > > ### Author Response · Authors · 2023-08-16
> > > **Response to Reviewer a5TN**
> > >
> > > Thank you for your acknowledgment of our work! We will carefully refine our paper for better clarity. Your willingness to re-evaluate the rating means a lot to us. If it's convenient, please take a moment to update the rating in the system. Do reach out if there are further queries or concerns!

---

### Official Review · Reviewer_5yZr · 2023-07-05

**Soundness:** 4 excellent
**Presentation:** 4 excellent
**Contribution:** 3 good
**Rating:** 8
**Confidence:** 3

**Summary:**

The paper presents LEPARD, a framework for reconstructing the 3D articulated shape of animals from a single in-the-wild image. It explicitly represents the parts as parameterized primitive surfaces (superquadrics) with global and local deformations in 3D. The authors employ a kinematics-inspired optimization to guide the deformations. Besides, LEPARD is trained solely using off-the-shelf deep features from DINO, without requiring any 2D or 3D annotations. Experiments on the Pascal-part and LASSIE datasets show the superiority of the proposed method.

**Strengths:**

- The paper is clearly written and easy to follow.
- The authors propose to use local non-rigid deformations to capture fine-grained shape details, which is different from previous works.
- The authors propose a framework to compute image-based forces based on the discrepancy of DINO features and the projected primitive parts.

**Weaknesses:**

There are a limited number of categories in the datasets evaluated. Can the authors qualitatively evaluate their methods on held-out images from other sources for the same training categories and some unseen categories (maybe fine-tuning a small set of images)?

**Questions:**

In L174, it is mentioned that "the estimated parameters for camera translation and rotation" are used. I am curious whether these camera parameters are trained by the image force only. Are there any additional supervisions or any missing implementation details (e.g., predicting multiple hypotheses as in A-CSM)? Because it can be quite ill-posed to optimize both camera parameters and part parameters.

**Limitations:**

The authors have not adequately addressed the limitations. It will be nice if the authors can discuss, e.g., 1) the effect of the quality of psuedo labels generated by DINO on the final performance; 2) the bottleneck of the current method for the further improvement.

---

> ### Author Rebuttal · Authors · 2023-08-09
>
> Thank you for your strong recognition of our work! We hope the following responses can address your concerns.
>
> >Q1: There are a limited number of categories in the datasets evaluated. Can the authors qualitatively evaluate their methods on held-out images from other sources for the same training categories and some unseen categories (maybe fine-tuning a small set of images)?
>
> A1: Thanks for the suggestion! We have tested on the Objaverse dataset for known categories such as elephants. The results are given in Fig. 2 of the Author Rebuttal PDF. Fine tuning on additional images for unknown categories might be out of the scope of this paper, and we will include it in future work.
>
> >Q2: In L174, it is mentioned that "the estimated parameters for camera translation and rotation" are used. I am curious whether these camera parameters are trained by the image force only. Are there any additional supervisions or any missing implementation details (e.g., predicting multiple hypotheses as in A-CSM)? Because it can be quite ill-posed to optimize both camera parameters and part parameters.
>
> A2: We do not use the multiple camera hypothesis approach from A-CSM [20] and U-CMR [14], but predict a single camera viewpoint similar to LASSIE [40] and Hi-LASSIE [41].
>
> >Q3: It will be nice if the authors can discuss the effect of the quality of pseudo labels generated by DINO on the final performance
>
> A3: Thanks for the suggestion! We added an experiment to test the effect of the DINO feature quality on the final performance. Specifically, we use the Pascal-Part dataset which provides GT masks for the evaluation of DINO feature quality. We sort the images based on the IoU between the DINO feature and the GT mask, and split these images into three groups according to their quality (IoU) rankings i.e., group 1, 2, 3 with the best, medium, and worst quality, respectively.  We then train a separate model on each group and evaluate their performance in terms of overall IoU and Part IoU. The average results over all three tested categories are reported in Table. 1 of the Author Rebuttal PDF. We observe that the performance of our approach only slightly degrades with lower DINO feature quality, which may also be due to the fewer training samples.
>
>
>
> >Q4: the bottleneck of the current method for further improvement.
>
> A4: The bottleneck is the performance of our method depends on the use of the correct number of parts for reconstruction. While this is not very restrictive, in future work we plan to extend our method so that it works without knowledge of the number of parts. Please see the general Author Rebuttal for details.

---

> > ### Comment · Reviewer_5yZr · 2023-08-13
> >
> > Thank the authors for the detailed reply and extra results. It shows that the model is quite robust. I would like to keep my rating.
> >
> > However, I still have a question about optimizing camera parameters in a self-supervised way. Is there a dataset bias (e.g., most animals face toward the camera so that only the range of azimuth is [-$\pi$, $\pi$]? Theoretically, without multiple hypotheses [1], the worst prediction error for a continuous function mapping $R^3$ to $SO(3)$ can be quite large.
> >
> > [1] Xiang, Sitao, and Hao Li. "Revisiting the continuity of rotation representations in neural networks." arXiv preprint arXiv:2006.06234 (2020).

---

> > > ### Author Response · Authors · 2023-08-16
> > > **Response to Reviewer 5yZr**
> > >
> > > Thank you for your feedback and acknowledgment of our work. In our implementation, we integrate the optimization of camera rotation parameters into the kinematics modeling, which provides more strict constraints and improves the performance of unsupervised learning. As suggested, incorporating multiple hypotheses could lead to even more robust optimization and will be included in our future exploration.

---

### Official Review · Reviewer_pQ1r · 2023-07-06

**Soundness:** 4 excellent
**Presentation:** 3 good
**Contribution:** 3 good
**Rating:** 7
**Confidence:** 4

**Summary:**

The paper describes a method for fitting K superquadric geometric primitives (enhanced with tapering, bending, and diffeomorphic local deformations) to a set of images of an animal category (e.g. elephant). The main contribution is that the method requires no supervision, and uses 2D feature correspondence to impose constraints on 3D shapes.

**Strengths:**

Superquadrics with limited deformations offer a good balance between geometric expressivity and parameter compactness.

The end-to-end image supervision of 3D geometry is fundamentally sound. Thus, relying only on 2D feature correspondences, and translating those to 3D constraints, is a robust way to aggregate information from an unstructured image collection.

The results are compelling and a clear improvement on previous work.

**Weaknesses:**

Animal bodies are kinematic chains, where one limb affects another. This approach does not take that in account.

Even though the paper focuses on animals, it would be quite useful to see how it performs on humans and how it compares to human-specific baselines.

The paper could use a section discussing the limitations and future work.

**Questions:**

Elephant ears and trunk seem part of the same head primitive in Figure 5. Yet the shape is quite complex. Is that due to local deformation? If so, then figure 6 should be changed to use the elephant example instead of the more minor tiger example.

How do you ensure that rotation matrices R are orthonormal with positive determinants?

How would you incorporate the fact that animals are a kinematic chain (bones connected to each other), rather than a bag of shape primitives?

Did you try running your method on humans? It would be very informative to see human results compared to human-specific methods.

The supplemental material shows interesting mappings between different animals. It would be impactful to show more of that analysis in the main paper. Are there any cross-category benchmarks to evaluate it on?


**Limitations:**

The paper does not explicitly address its limitations in a traditional limitations section.

It is not explicitly stated whether the number of primitives, K, is computed automatically or manually specified. It would be a limitation if it has to be manually specified.

The method treats each primitive separately from others, which does not faithfully represent the kinematic chain that is an animal shape.

---

> ### Author Rebuttal · Authors · 2023-08-09
>
> Dear reviewer, we greatly appreciate your recognition of our work, and thank you for your valuable comments! We hope the following responses can address your concerns.
>
> >Q1: Animal bodies are kinematic chains, where one limb affects another. This approach does not take that into account. How would you incorporate the fact that animals are a kinematic chain (bones connected to each other), rather than a bag of shape primitives?
>
> A1: LEPARD is a general approach that is not limited to animal bodies and can also be applied to a wide range of objects such as inanimate (human-made, natural objects), and animate (animals, humans) objects. While in this paper, we focus on animals, however, our approach can be extended in future work to add kinematic constraints and other types of objects. In future work we plan to extend our approach to create a kinematically constrained primitive-based model of an animal or human, and fit to the target shape data (static or dynamic).
>
> >Q2: Performance on humans.
>
> A2: In this paper we focus on 3D part discovery of articulated animal shapes from 2D images and mainly compare with those methods which work on the same task. In future work we plan to use our method on humans due to its general representation ability.
>
> >Q3: Elephant ears and trunk seem part of the same head primitive in Figure 5. Yet the shape is quite complex. Is that due to local deformation? If so, then Figure 6 should be changed to use the elephant example instead of the more minor tiger example.
>
> A3: The shape of ears and trunk of an elephant are reconstructed due to a combination of tapering, bending (i.e., global deformations) as well as local deformations. In Fig.5 of the main paper we ablate using the shape of tigers to better demonstrate the effect of local deformations.
>
> >Q4: How do you ensure that rotation matrices R are orthonormal with positive determinants?
>
> A4: We predict 3D rotations as normalized 4D unit quaternions, which are converted to valid rotation matrices (that are orthonormal with positive determinants) using a closed-form transformation matrix.
>
> >Q5: The supplemental material shows interesting mappings between different animals. It would be impactful to show more of that analysis in the main paper. Are there any cross-category benchmarks to evaluate it on?
>
> A5: To the best of our knowledge, there are no public cross-category benchmarks to show such 3D part level consistency across species. However, we will add sufficient qualitative comparisons for this to the main paper/supplementary.

---

> > ### Comment · Reviewer_pQ1r · 2023-08-17
> > **Thanks for the rebuttal**
> >
> > My rating remains.
> >
> > A further comment on A1:
> >   You cannot claim "LEPARD is a general approach that is not limited to animal bodies" if you only demonstrate it on animal bodies.

---

> > > ### Author Response · Authors · 2023-08-19
> > > **Response to Reviewer pQ1r**
> > >
> > > Thank you for your feedback and additional comment.
> > >
> > > Our intention was to emphasize the foundational principles of LEPARD, which are designed to be generalizable. However, we understand that, without empirical evidence demonstrating its efficacy beyond animal bodies, such a claim might be premature. We will ensure that the revised manuscript accurately reflects the scope of LEPARD based on the data presented. In the future, we will actively work on diversifying our dataset and testing LEPARD on a broader range of object categories to substantiate its general applicability.

---

### Official Review · Reviewer_ifgQ · 2023-07-06

**Soundness:** 4 excellent
**Presentation:** 3 good
**Contribution:** 3 good
**Rating:** 6
**Confidence:** 4

**Summary:**

The paper proposes a part-based method to reconstruct 3D shapes in a category-specific manner. Compare against its baseline LASSIE, the paper uses an elegant primitive part representation that could capture both global and local deformations to increase the fidelity of reconstruction. The method also does not need test-time optimization process like LASSIE. The proposed method outperforms LASSIE quantitatively and qualitatively.

**Strengths:**

1. The proposed primitive part representation is novel. Compare against modeling deformation implicitly,  the parameterization of the representation is compact and intuitive for understanding. The new representation increase the fidelity of the reconstruction.

2. No need for test-time processing.  The proposed method does not need per-instance optimization and only requires a forward-pass.

3. No 3D or input requirements. The method takes only image as input, without the need of mesh template, category skeleton as additional input. For training, the method is self-supervised and does not need 2D/3D annotations and could be easily scale-up to in-the-wild data.

4. Impressive qualitative results. The visual result shown in Fig. 4 is impressive. The results looks quite strong as the detailed structure of the animal could be fully recovered in a primitive-based approach.

5. Outperform baseline method in all categories.

**Weaknesses:**

1. The motivation of introducing image force in the training objective is not well explained. As LASSIE adopts simple silhouette loss by differentiable rendering, the paper introduces complex Jacobian matrices and generalized forces. What kind of benefit could we get from this and what is the difference and limitation of using silhouette loss against LASSIE. The author should include more experiments to demonstrate the superiority of such kinematic-inspired procedure.

2. Several questions not fully understand. Please questions below.

3. No discussion of the limitation of the method and when the method would fail.

**Questions:**

1. How does the training objective guarantee semantic consistency. The paper adopts the same process of DINO features as LASSIE used for training. However, unlike LASSIE, the method does not need test-time optimization and the training objective does not explicitly model 2D-3D consistency. The author should explain how they achieve better results than LASSIE in more details.

2. How to choose number of parts for each category? Different from LASSIE, the method does not take skeleton as input, so how to choose number of parts for unknown category. If the number of parts is high, will there be over-segment problem and lose the semantic meaning and consistency?

**Limitations:**

Please see comments above. It would also be quite interesting if the output parts could be further retargeted to novel poses. The part-based approach is hard for animation against parametric models.

---

> ### Author Rebuttal · Authors · 2023-08-09
>
> Dear reviewer, we greatly appreciate your recognition of our work, and thank you for your valuable comments! We hope the following responses can address your concerns.
>
> >Q1: The motivation of introducing image force in the training objective is not well explained. As LASSIE adopts simple silhouette loss by differentiable rendering, the paper introduces complex Jacobian matrices and generalized forces. What kind of benefit could we get from this and what is the difference and limitation of using silhouette loss against LASSIE. The author should include more experiments to demonstrate the superiority of such kinematic-inspired procedures.
>
> A1: The image force is calculated using the pixel-level difference between the primitive parts and the 2D image, and measures how well the primitive surface needs to be deformed to match the ground truth shape in 2D. The image force can be viewed the same as the silhouette loss in LASSIE, which is obviously not enough for both approaches, especially in the scenario of 2D-3D reconstruction. As opposed to LASSIE which uses multiple separate regularization terms resulting in sub-optimal performance, we calculate image forces in 2D and convert them into generalized forces to enable a joint regularization of the shape-related parameters. This allows us to regularize each sub-transformation of each primitive part deformation, and provides more effective constraints during training without requiring knowledge of joint locations. The extensive experiments demonstrate the effectiveness and robustness of our optimization strategy. For comparison, in Table. 2 of the Author Rebuttal PDF, we provide an ablation study using only the image forces in the loss function and demonstrate consistent superior LEPARD performance against baselines.
>
> >Q2: How does the training objective guarantee semantic consistency. The paper adopts the same process of DINO features as LASSIE used for training. However, unlike LASSIE, the method does not need test-time optimization and the training objective does not explicitly model 2D-3D consistency. The author should explain how they achieve better results than LASSIE in more detail.
>
> A2: See also A1 for details on how our method works which is different from LASSIE. We use kinematic-based modeling which jointly constrains the parameters of translation, rotation, and deformations and results in a more robust solution. As opposed to LASSIE, these parameters are learnt and guide each primitive to deform and fit more accurately to a part of the shape. The kinematic constraints result in learning a more consistent parametric transformation among similar animal categories.
>
> >Q3: It would also be quite interesting if the output parts could be further retargeted to novel poses. The part-based approach is hard for animation against parametric models.
>
> A3: Our model is parametric (i.e., each part is fully parameterized by a few shape-related parameters) and thus is naturally suitable for reposing by changing the translation or rotation matrix. Note that since we use fully explicit representation, we can even change the bending or tapering degree of each part by adjusting the global deformation parameters. This is impossible to do with implicitly represented parts as in LASSIE and the other baseline methods since they use MLPs to directly estimate the point-wise deformation field. In addition to being more robust to missing data or gaps in the data compared to implicit methods, our explicit shape representation approach offers shape explainability (e.g., tapering, bending shape information).

---

> > ### Comment · Reviewer_ifgQ · 2023-08-16
> >
> > Thank for the detailed response by the author. The rebuttal address most of my concerns. Here are my remaining concerns and comments:
> >
> > 1. The image force is better than silhouette  constraint from Tab 2 in the rebuttal. Please consider add this exp to the future version.
> >
> >
> > 2. It is encouraged to add the retargeting visualization results as explained in the rebuttal. This could be a clear advantage compare against other implicit representations.  It would be useful and interesting to see how it work by changing some bending and tapering degrees.
> >
> >
> > 3. Regarding limitation, I am a bit confused why only 16 parts work, what will happen if we increase/decrease the number of parts. For a more complicated shape, does 16 part sufficient. For a simple shape, does 16 parts make it over-complicated. The limitation should be included and presented more clearly in the latest version.

---

> > > ### Author Response · Authors · 2023-08-19
> > > **Response to Reviewer ifgQ**
> > >
> > > Thank you for your further insights and feedback. Here is our response to the comments:
> > >
> > > 1. We acknowledge your suggestion regarding the performance of the image force compared to the silhouette constraint, as observed from Table 2 in our rebuttal. In the subsequent version of our work, we will ensure that this comparison is explicitly included.
> > >
> > > 2. We agree that this can be a great addition, showcasing the practical advantages of our approach compared to other implicit representations. We will incorporate visualizations that demonstrate the modifications to bending and tapering degrees to provide a clearer understanding of our method's efficacy in the context of shape manipulations.
> > >
> > > 3. For comparison we use the same number of parts as the baseline method. The choice of 16 parts was derived from the anatomical understanding of four-limb animals, capturing the core structure while preserving semantic meaning across different species, despite their shape variations. For object categories other than four-limb animals, the number of parts could be different. The determination of the optimal number of parts will be contingent on the anatomical specifics of the object category in question. We'll investigate this further and provide a more detailed explanation in the updated version. This will give readers a better insight into how the number of parts affects the method's performance for various shapes, as well as any computational considerations.

---

### Author Rebuttal · Authors · 2023-08-09

Dear AC and reviewers, we are grateful for the strong recognition and valuable comments of our work. In the following, we will first answer the common concern of all reviewers, followed by answers to each reviewer's comments.

The common concern is mainly the limitation of our approach.

We share the same assumption as LASSIE that all the four-limb animals share the same 3D part structure despite considerable shape variations across species, which allows LASSIE and LEPARD to apply the same number of parts to all four-limb animals. The only difference is that LASSIE uses more strict supervision, i.e., the number of joints K+1 (which determines the number of parts K) and their locations. As opposed to LASSIE, we use the less restrictive common knowledge K=16 for parts to all four-limb animals and we do not define or use the locations of the K (of the K+1 joints). We use kinematic-based modeling which allows us to discover the K parts. In addition, note that the number of parts is a common assumption for unsupervised 3D part reconstruction from a single image[1-5].  The experiments in the paper demonstrate the semantic consistency and reconstruction accuracy of LEPARD, which shows that our method works well in real-world scenarios/images.

One potential limitation is that if we use other than K=16 for the number of parts, the performance may degrade because only up to 16 parts are visible in the images. For the same reason the performance of LASSIE may degrade if they change the number of joints that corresponds to K=16 parts. Our future work will include active part discovery by monitoring motion changes to overcome the need for initializing the number of parts.


[1] Tulsiani, Shubham, et al. "Learning shape abstractions by assembling volumetric primitives." Proceedings of the IEEE Conference on Computer Vision and Pattern Recognition. 2017.

[2] Paschalidou, Despoina, Ali Osman Ulusoy, and Andreas Geiger. "Superquadrics revisited: Learning 3d shape parsing beyond cuboids." Proceedings of the IEEE/CVF Conference on Computer Vision and Pattern Recognition. 2019.

[3] Deng, Boyang, et al. "Cvxnet: Learnable convex decomposition." Proceedings of the IEEE/CVF Conference on Computer Vision and Pattern Recognition. 2020.

[4] Paschalidou, Despoina, et al. "Neural parts: Learning expressive 3d shape abstractions with invertible neural networks." Proceedings of the IEEE/CVF Conference on Computer Vision and Pattern Recognition. 2021.

[5] Tertikas, Konstantinos, et al. "Generating Part-Aware Editable 3D Shapes Without 3D Supervision." Proceedings of the IEEE/CVF Conference on Computer Vision and Pattern Recognition. 2023.

---

### Decision · Program_Chairs · 2023-09-21

**Decision:**

Accept (poster)

**Comment:**

This is a technically solid paper which introduces a novel part representation for unsupervised part-based 3D reconstruction. The qualitative and quantitative results show clear benefits over prior work using more restrictive representations while also removing the need for test-time optimizations. Although there were concerns regarding the limited applications to quadrupeds, this is the typical setting considered across prior works.

The reviewers all recommend (clear/weak/borderline) acceptance (including Reviewer XxbL based on their subjective comments), and the AC agrees. However, the authors are strongly encouraged to improve the presentation of the technical approach considering a broader audience in mind.